# Oxidative Strong Metal–Support Interactions

Xiaorui Du [1], Hailian Tang [2] and Botao Qiao [1,3,*]

1    CAS Key Laboratory of Science and Technology on Applied Catalysis, Dalian Institute of Chemical Physics, Chinese Academy of Sciences, Dalian 116023, China; duxiaorui@dicp.ac.cn
2    College of Chemistry and Environmental Science, Hebei University, Baoding 071002, China; tanghl_soso@126.com
3    Dalian National Laboratory for Clean Energy, Dalian Institute of Chemical Physics, Chinese Academy of Sciences, Dalian 116023, China
*    Correspondence: bqiao@dicp.ac.cn

**Abstract:** The discoveries and development of the oxidative strong metal–support interaction (OMSI) phenomena in recent years not only promote new and deeper understanding of strong metal–support interaction (SMSI) but also open an alternative way to develop supported heterogeneous catalysts with better performance. In this review, the brief history as well as the definition of OMSI and its difference from classical SMSI are described. The identification of OMSI and the corresponding characterization methods are expounded. Furthermore, the application of OMSI in enhancing catalyst performance, and the influence of OMSI in inspiring discoveries of new types of SMSI are discussed. Finally, a brief summary is presented and some prospects are proposed.

**Keywords:** oxidative strong metal–support interactions; strong metal–support interaction; supported metal catalyst; metal-oxide interfaces; charge transfer; heterogeneous catalysis

## 1. Introduction

Supported metal catalysts, in which nano-scale active metals are dispersed on a high surface area support, are the most widely used and extensively studied heterogeneous catalysts, and play an irreplaceable role in the modern chemical industry [1–4]. Initially, the primary role of the support was regarded as to disperse and stabilize active metal components, as well as enhance the mechanical strength and heat resistance of the catalyst [5,6]. However, with further understanding of the structure–activity relationship, the decisive role of the support in determining the catalyst performance was soon realized. Beyond the abovementioned functions, the chemical interactions between the active metal and the support were found to significantly affect the catalyst properties and had long been focused as an effective route to improve catalysts [1,5,7–14].

One of the most far-reaching discoveries that highlights these interactions is an unusual phenomenon discovered by Tauster et al. in the late 1970s, that $TiO_2$-supported Pt-group metals will lose their capability to adsorb small molecules (such as CO and $H_2$) following high-temperature reduction [15,16]. They named this phenomenon the strong metal–support interaction (SMSI), and subsequently extensive researches have been devoted to uncover its nature and performances [17–31]. By the late 1980s, based on the catalyst systems of the reducible oxide (mainly of groups IIA-VB)-supported Pt-group metals, the main characteristics of SMSI induced by high-temperature reduction have been confirmed consecutively, involving: (1) striking suppression of small-molecule adsorption on metal; (2) mass transport that metal particles will be encapsulated by the reduced support; (3) electron transfer from the support to metal; (4) a reversal of the above phenomena following re-oxidation treatment. Such high-temperature reduction-induced SMSI is called classical SMSI, and the four features are also criteria for identifying its occurrence. These features endow the catalysts with great tunability for their structure and properties, thus

induction of SMSI has developed as an effective way to modify catalyst performance for diverse reactions.

Classical SMSI between Pt-group metals and reducible oxides has continuously been studied in both catalysis application and surface science for the next three decades, and the understanding on it has progressed significantly. Intermetallic bonding [16,17], electron transfer [29–34], and mass transport between metal particles and the support [21–23,34–38] were all proposed as the origination of SMSI. However, no consensus has been reached so far. Meanwhile, the minimization of surface energy was considered as a driving force for the encapsulation reaction in SMSI, and metals with higher surface energies and large work functions were regarded to be indispensable [33,39–41]. Based on the classical SMSI theory, the SMSI-active catalyst system was limited, as shown in Figure 1 [33], in which only the metals in region I would be able to form SMSI with TiO$_2$. Au (and IB group metals), with a relatively lower work function and surface energy respect to Pt-group metals, had thus been regarded as SMSI-inert for a long time, which is also due to the lack of direct evidence for SMSI between Au and TiO$_2$ in surface science studies [38,39,42,43].

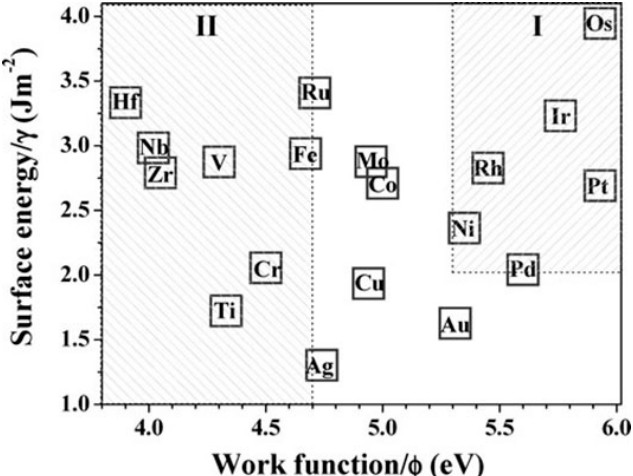

**Figure 1.** Relationship between surface energy ($\gamma_M$) and work function ($\varphi$) of different transition metals. Reproduced with permission from [33]. Copyright: 2005 by the American Chemical Society.

A breakthrough came from Mou's group in 2012 [44]. They found an oxygen-induced SMSI phenomenon in an Au/ZnO-nanorod catalyst, that, following oxidation under 300 °C, the Au nanoparticles will be encapsulated by ZnO accompanied by electron transfer from Au to the support, which will be reversed by hydrogen treatment. This work, for the first time, not only extended the conditions for evoking SMSI but also opened a prelude for the study of SMSI in Au-based catalysts. Only about four years later, a similar oxidative SMSI phenomenon was revealed by Tang et al. in nonoxide (hydroxyapatite and phosphate)-supported Au catalysts, expanding the territory of SMSI to the non-oxide support system [45]. Furthermore, in 2018 [46], they confirmed that the nonoxide- and ZnO-supported Pt-group metals are also applicable for this oxygen-induced SMSI, and formally proposed the concept of "oxidative strong metal–support interactions (OMSI)" to distinguish from classical SMSI, which was triggered by reductive conditions. It has been proved that controlled OMSI is effective in tuning the catalyst performances [44–47], and a few discoveries inspired by OMSI were consecutively reported thereafter, such as uncovering the classical SMSI between Au (actually the IB group metals) and TiO$_2$, and SMSI between Au and layered double hydroxide support [48,49].

OMSI has not only shed new light on the understanding of SMSI phenomena but also provided new opportunities for the design and development of high-performance catalysts, thus becoming an important part of the generalized SMSI field. This review focuses on the current understanding and unsolved problems on OMSI which may be instructive for the further development of OMSI. The major content includes the definition

of OMSI and its difference from classical SMSI, the characterization and identification of OMSI, and the application and influence of OMSI. Finally, a brief summary and perspective is presented. We hope this review can raise interest for researchers and inspire further attempts to explore more potential of OMSI in heterogeneous catalysis.

## 2. Definition, Features, and Catalyst Systems of OMSI

OMSI can be defined as a phenomenon occurring in a supported metal catalyst that is triggered by oxidative (or non-reductive) conditions with the typical features resembling that of SMSI. Current OMSI systems involve ZnO nanorod-supported Au, nonoxide (hydroxyapatite and phosphate)-supported Au and Pt-group metals, and ZnO-supported Pt-group metals.

Initially, it was found that Au nanoparticles supported on the ZnO-nanorod (Figure 2a) will be encapsulated by the support after it is treated at 300 °C under an oxygen atmosphere, accompanied by the decrease in CO adsorption capacity and the electron transfer from the surface of Au nanoparticles to ZnO, which will be all reversed when the treated catalyst is reduced under $H_2$ [44]. These typical characteristics are analogous to that of classical SMSI. Further, it was revealed in hydroxyapatite (HAP)- and $LaPO_4$-supported Au catalysts (Figure 2b) that upon calcination in the oxygen atmosphere at ≥300 °C, the surface of Au nanoparticles will be covered by a HAP thin layer and completely encapsulated at 600 °C, and correspondingly, the CO adsorption gradually decreased to disappeared [45]. Following a reduction at 500 °C, the overlayer retreated and the CO adsorption recovered. Reversible electron transfer between Au nanoparticles and HAP was also evidenced. Subsequently, the same phenomenon also triggered by high-temperature oxidation was confirmed in HAP- and ZnO-supported Pt-group metals (Figure 2c) [46]. These successive discoveries highlighted the universality of the OMSI effect, and thus developed to become an individual concept. The common characteristics of OMSI evoked by high-temperature oxidation (or an inert atmosphere) can be summarized as:

(1) Small-molecule of CO or $H_2$ adsorption on metal will be significantly suppressed;
(2) Mass transport that the support would encapsulate metal particles;
(3) Electron transfer from metal to the support resulting in a positively charged metal species;
(4) A reversal of the above phenomena following reduction treatment.

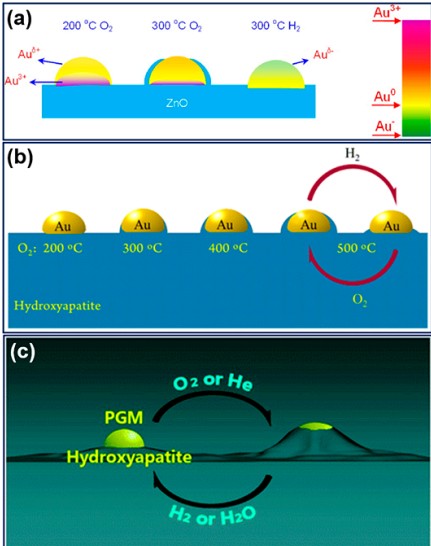

**Figure 2.** Schematic illustration of OMSI for (**a**) Au/ZnO-nanorod, reproduced with permission from [44], copyright: 2012 by the American Chemical Society, (**b**) Au/HAP, reproduced with permission from [45], copyright: 2016 by the American Chemical Society, and (**c**) HAP supported Pt-group metals, reproduced with permission from [46], copyright: 2018 by The Royal Society of Chemistry.

The comparison of the inducing conditions and main characteristics of OMSI and classical SMSI was listed in Table 1.

**Table 1.** Comparison between OMSI and classical SMSI.

|  | Classical SMSI | OMSI |
| --- | --- | --- |
| typical catalyst system | reducible oxide-supported Au or Pt-group metals | HAP- or ZnO-supported Au or Pt-group metals |
| inducing conditions | high-temperature reduction | high-temperature oxidation |
| suppression of adsorption | yes | yes |
| mass transport (encapsulation) | yes | yes |
| electron transfer | support to metal | metal to support |
| reversibility | yes | yes |

## 3. Identification and Characterization of OMSI

The abovementioned four typical features are also the criteria for the identification of OMSI occurrence, which generally needs a combination of multiple characterization methods to confirm. In this section, the commonly used characterization methods for OMSI research are outlined according to the aspects of adsorption behavior, mass transport, and electron transfer.

### 3.1. Adsorption Behavior

Quantitative chemisorption measurement is conventionally used to characterize the adsorption capacity of metal catalysts and is the earliest employed to verify the occurrence of SMSI [15,16,18]. With the development of modern spectroscopic techniques with improved sensitivity and efficiency, chemisorption measurement has gradually been replaced, especially for the qualitative characterization of the catalyst adsorption performance. One of the most typical spectroscopic techniques is the in situ molecular probe infrared (IR) spectroscopy. Due to site-specific sensitivity in detecting the adsorption property of metal surface, it has been popularized in heterogeneous catalysis and become an indispensable approach to measure adsorption behavior of catalysts in the SMSI study [28,44–50]. Tang et al. carried out an in situ diffuse reflectance infrared Fourier transform spectroscopy (DRIFTS) measurement of CO adsorption on Au/HAP treated by high-temperature oxidation under different temperatures, an instructive case for the application of IR spectroscopy to the identification of OMSI [45]. As shown in Figure 3a, the bands at 2102~2110 $cm^{-1}$ were ascribed to the CO adsorbed on the Au surface (CO-Au). Obviously, with increasing calcination temperature, the intensity of the CO-Au bands decreased gradually, and disappeared after being calcined at 600 °C, indicating the suppression of small-molecule adsorption on Au following high-temperature oxidation. After subsequent reduction under $H_2$ at 500 °C (Au/H-500-$H_2$), the CO-Au band recovered. These are strong evidences to affirm the occurrence of OMSI. Meanwhile, the blue-shift of CO-Au from 2102 to 2110 $cm^{-1}$ indicated the generation of positively charged Au species with increasing calcination temperature, which can be a proof for the electron transfer from Au nanoparticles to HAP. Similar measurements were further carried out to prove the reversible suppression of chemisorption on Pd/HAP and Pt/HAP by high-temperature oxidation/reduction, confirming the occurrence of OMSI in Pt-group metal-based catalysts (Figure 3b,c) [46]. It should be noted that for Pt-group metal-based catalysts, generally high-temperature oxidation will cause relatively high metal valence, which would result in a weaker IR response for the CO adsorption, thus potentially causing confusion in identifying OMSI. Control experiments thus will be needed to confirm whether the CO adsorption on oxidized metal species can be detected [46].

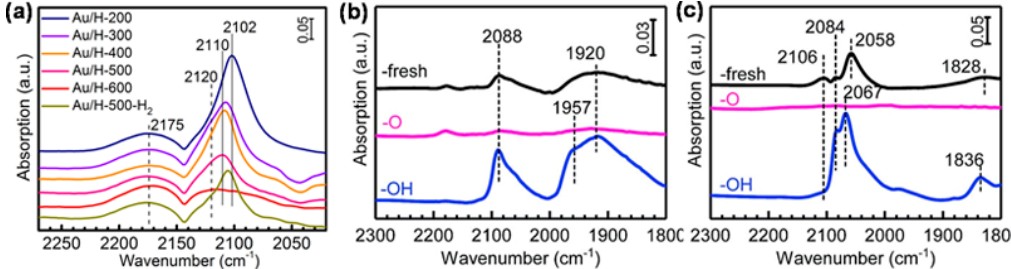

**Figure 3.** (**a**) In situ DRIFT spectra of CO adsorption on Au/H-X (H: HAP; X: the calcination temperature) and Au/H-500-H$_2$ (obtained by further reducing Au/H-500 at 500 °C); the bands at 2175 and 2120 cm$^{-1}$ can be assigned to the gas CO. Reproduced with permission from [45], copyright: 2016 by the American Chemical Society. (**b**,**c**) In situ DRIFT spectra of CO adsorption on (**b**) Pd/HAP and (**c**) Pt/HAP samples. "-fresh" represents the freshly synthesized samples without calcination, "-O" represents samples calcined under 10 vol% O$_2$/He flow at 500 °C, "-OH" represents samples further reduced under 10 vol% H$_2$/He flow at 250 or 500 °C for Pd/HAP and Pt/HAP, respectively. Reproduced with permission from [46], copyright: 2018 by The Royal Society of Chemistry.

### *3.2. Mass Transport*

Mass transport in OMSI mainly refers to the reversible encapsulation of metal by the deformed support, which generally can be intuitively observed by high resolution transmission electron microscopy (HRTEM). In 1985, Singh et al. directly observed a thin overlayer on the Rh surface through HRTEM in Rh/TiO$_2$ where SMSI occurred, thus inferring that the overlayer inhibited the adsorption of small molecules on the catalyst [23]. Since then, the SMSI-provoked encapsulation phenomenon has been found in various catalysts, becoming an essential basis for identifying the occurrence of SMSI, and HRTEM thus became indispensable [11,26,27,37,51,52].

Migration of ZnO onto the top surface of Au nanoparticles was observed after treating the Au/ZnO-nanorod in oxygen at 300 °C, which directly led to the discovery of OMSI [44]. As shown in Figure 4a, the source of the encapsulation layer was determined by measuring the lattice spacing of the overlayer (0.26 nm) that is in accord with the (002) plane of ZnO. It was also clear that after being treated under the H$_2$ atmosphere, the ZnO thin layer retreated (Figure 4b). By HRTEM, Tang et al. found that the coverage of HAP on Au nanoparticles depended on the calcination temperature [45]. As shown in Figure 5a–e, the encapsulation layer on the surface of Au nanoparticle appeared when the Au/HAP calcined at 300 °C, then the layer gradually spread as the calcination temperature increased, and finally completely wrapped the Au nanoparticle after being calcined at 600 °C. By subsequent treatment under pure H$_2$ at 500 °C, the encapsulation layer retreated (Figure 5f). It should be noted that for the ultrasmall metal nanoparticles, using HRTEM is difficult to identify the thin cover layer, thus a larger nanoparticle size of the catalyst is required sometimes to confirm OMSI [46]. Meanwhile, the difference in contrast between metal nanoparticles and the support will also affect the observation results. Therefore, the popularization of the electron microscopy technique with both ultrahigh definition and surface sensitivity is of importance for the in-depth exploration of OMSI.

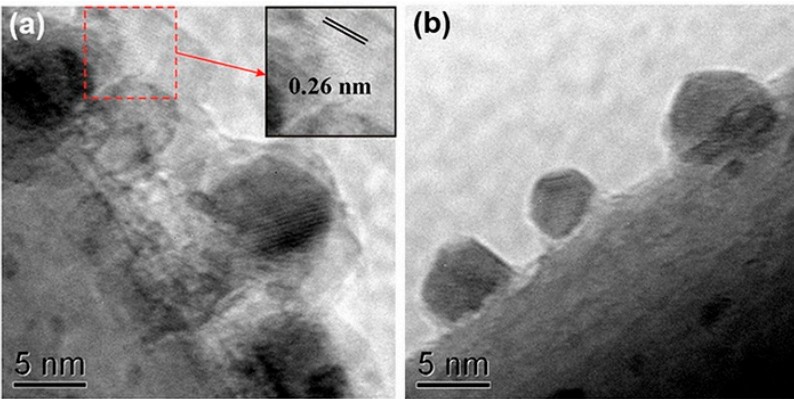

**Figure 4.** HRTEM images of (**a**) the Au/ZnO-nanorod catalyst pretreated at 300 °C in 10% $O_2$/He and (**b**) Au/ZnO-nanorod sample further reduced at 300 °C in $H_2$ after oxidation pretreatment. Reproduced with permission from [44], copyright: 2012 by the American Chemical Society.

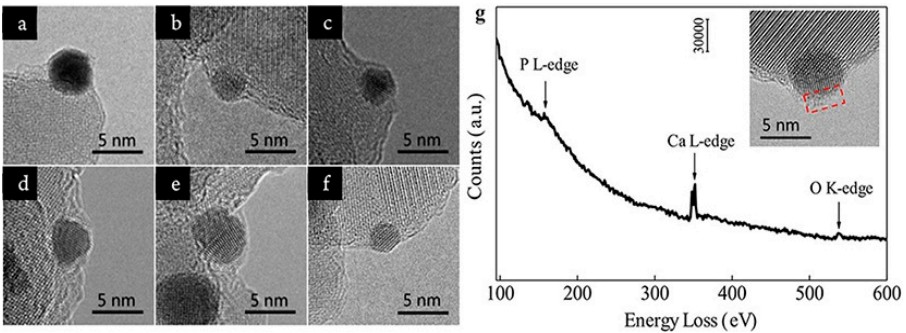

**Figure 5.** (**a**–**e**) HRTEM images of Au/H-X samples where "H" represents HAP and "X" represents the calcination temperature: (**a**) Au/H-200, (**b**) Au/H-300, (**c**) Au/H-400, (**d**) Au/H-500, and (**e**) Au/H-600; (**f**) HRTEM image of sample obtained by further reducing Au/H-500 at 500 °C under $H_2$. (**g**) The electron energy loss spectrum (EELS) spectrum of Au/H-600 where the Au nanoparticles were completely encapsulated. Reproduced with permission from [45], copyright: 2016 by the American Chemical Society.

Verifying the composition of the encapsulation layer is critical to concrete the mass transport of the support. For most SMSI or OMSI systems, the encapsulation layer is generally amorphous, which could not be determined by measuring the lattice spacing. The electron energy loss spectrum (EELS) is powerful to analyze the composition and the element valence of materials, thus has long been used in SMSI studies [53–55]. For instance, the cover layer on Au nanoparticles in the Au/HAP calcined at 600 °C was detected by EELS, and the result (Figure 5g) showed the layer was composed of P, Ca, and O, confirming it was derived from the HAP support [45].

Migration of the support onto the surface of metal nanoparticle during OMSI is a dynamic process which occurred at the metal/support interface and may cause structural changes in the metal nanoparticle. Real-time monitoring of this dynamic process is inevitably crucial to reveal the OMSI mechanism and the reconstructed interface structure, which requires advanced (in situ) environmental TEM and enabled the timely observation under a different atmosphere. A state-of-the-art in situ TEM study has been performed on Pd/TiO$_2$, realizing an atomic description of the classical SMSI phenomenon [56]. However, related research on the OMSI system has not been reported yet, which is highly anticipated.

*3.3. Electron Transfer*

Generally, the electron transfer during OMSI is mainly manifested by the perturbation in the valence state and the change in electronic structure of the supported metal, which can be examined by element-specific spectroscopic techniques that are sensitive to chemical

state and atomic structure. Earlier for the research of classical SMSI, X-ray photoelectron spectroscopy (XPS), ultraviolet photoelectron spectroscopy (UPS), and Auger electron spectroscopy (AES) have been employed in characterizing the high-temperature reduction-induced chemical state changes of metal [20,22,25,36,53,54,57,58]. The aforementioned CO-probed IR spectroscopy can also be used to testify the phenomenon of electron transfer as it is sensitive to the metal electronic structure. On the other hand, the electron transfer that accompanied by the support deformation at interface will inevitably impact the surface local coordination structure of the support. For the oxide support-based catalysts, the SMSI- or OMSI-induced electron perturbation can be verified by electron paramagnetic resonance (EPR), a technique sensitive to species with unpaired electrons such as oxygen vacancies or metal ions in paramagnetic valence states [59–62]. The metal atom-centered local coordination structure that strongly depends on the interaction between metal and the support can be examined by X-ray absorption spectroscopy (XAS), which is also necessary in SMSI and OMSI studies [57,63,64].

The above characterization techniques are available for most of the SMSI or OMSI catalysts, and appropriate combination of the techniques is generally required to validate the existence of electronic perturbation. A comprehensive study on Au/ZnO-nanorod catalysts has been carried out by Mou's group by combining in situ DRIFTS of CO adsorption, EPR, XAS, and XPS, which demonstrated the electron transfer and local structural changes that are evoked by OMSI [44]. The in situ DRIFTS of the CO adsorption result is shown in Figure 6a, where the CO-Au band blue-shifted from 2101 to 2113 $cm^{-1}$, implying the Au nanoparticles' surfaces were positively charged with increasing oxidation temperature. Following subsequent heat treatment under $H_2$, the CO-Au band red-shifted to 2108 $cm^{-1}$, and a band at 2048 $cm^{-1}$ that was attributed to the CO adsorbed on a negatively charged Au surface appeared. These clearly manifested the reversible electron transfer between Au nanoparticles and ZnO, which was further confirmed by EPR. As shown in Figure 6b, the singly ionized oxygen vacancy (g = 1.960) of ZnO converted to be paramagnetically silent by obtaining an electron from Au after loading Au nanoparticles and oxidative treatment. Following further treatment under $H_2$, the signal appeared again due to a regain of unpaired electrons by losing an electron. Furthermore, XAS, including extended X-ray absorption fine structure (EXAFS) and X-ray absorption near edge structure (XANES) spectra, were measured to illustrate the coordination environment and valence state of Au in the Au/ZnO-nanorod with OMSI evoked (Figure 6c,d). By detailed analysis of the EXAFS data, an Au-O-Zn interaction at the interfaces was found in the samples oxidized at 60, 120, and 200 °C, which disappeared when calcined at 300 °C. After further reduction at 300 °C, a peak ascribed to the AuZn alloy-like state can be detected (peak 2 in Figure 6c) and the corresponding XANES spectra (Figure 6d) were similar with that of Au foil, which means the *d* orbital of Au was almost fully occupied. The retreat of the encapsulation ZnO layer on the Au nanoparticle was thus inferred to be due to the formation of AuZn alloy following reductive heat treatment. Finally, XPS was employed to examine the valence state of Au, that Au (III) species were detected in the oxidized sample and then extinct by subsequent reduction. It was inspired from the above results that the interfacial electronic structure of the catalyst will be strongly affected by OMSI, which provide potential opportunities for tuning catalytic performance.

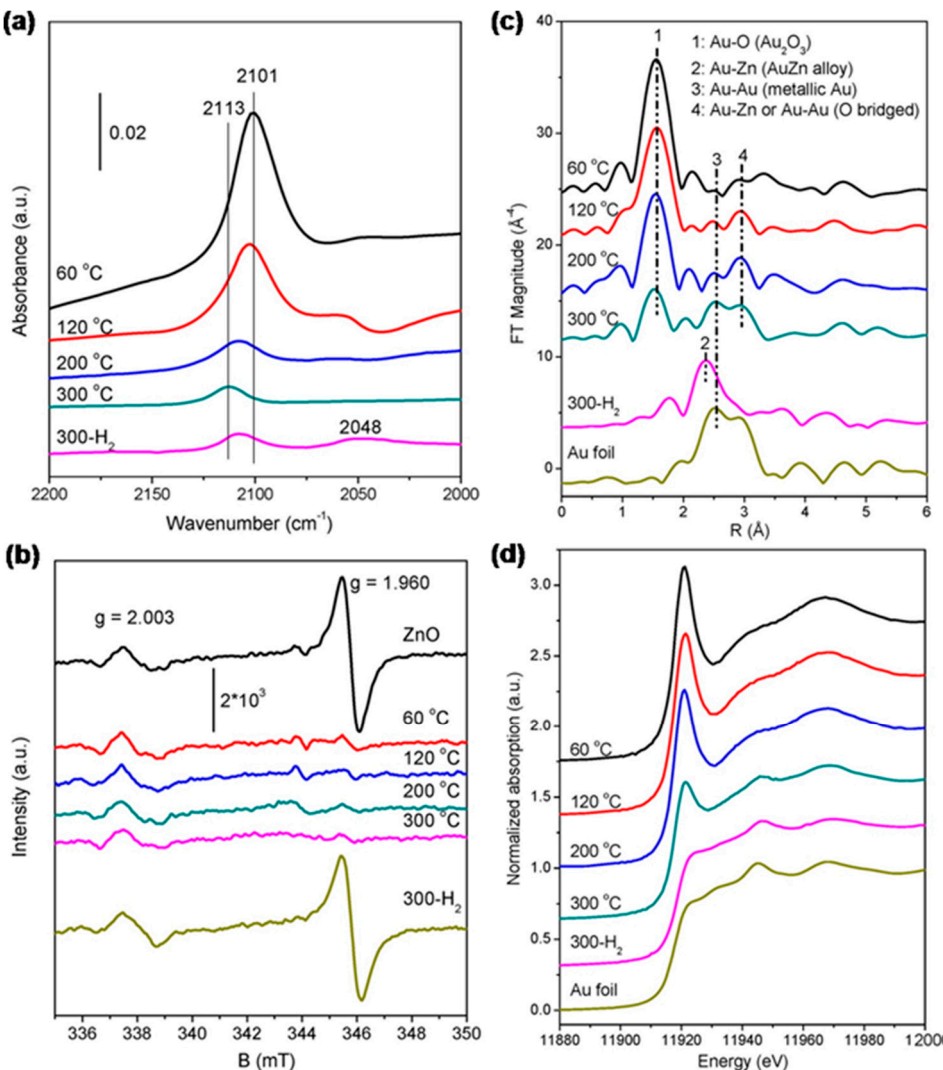

**Figure 6.** (**a**) In situ DRIFT spectra of adsorbed CO on Au/ZnO-nanorod catalysts; (**b**) EPR spectra of Au/ZnO-nanorod catalysts; (**c**) Fourier transform of $k^3$-weighted EXAFS spectra (without phase correction) and (**d**) normalized XANES spectra at the Au $L_{III}$-edge of 20Au/ZnO-nanorod catalyst. The catalysts were calcinedat 60, 120, 200, and 300 °C and then reduced at 300 °C. Reproduced with permission from [44], copyright: 2012 by the American Chemical Society.

## 4. Application and Influence of OMSI

The SMSI effect is able to alter the catalyst properties of the adsorption capacity, interface structure, and the electronic state, thus it has been extensively applied to enhance the catalyst performance for diverse reactions and recognized to be a highly effective method currently [18,50,65–75]. Admittedly, realizing the appropriate state of SMSI for different reactions is critical and difficult, which is limited by the precondition for the initiation and maintenance of SMSI. For instance, high-temperature reduction or a reductive atmosphere is prerequisite for classical SMSI, therefore the application of classical SMSI in non-reducing conditions is constrained. On this premise, discovery of the OMSI effect will enrich the potential application of the metal–support interaction under specific conditions to a large extent. On the other hand, OMSI truly broadens the territory of metal–support interaction, inspiring more explorations on similar phenomena that occurred in catalysts with multiple metals and supports and evoked by various conditions. This is of great value in in-depth cognition of metal–support interaction and catalytic mechanism. In this section, the latest progresses of OMSI application in enhancing catalyst performance, and recent breakthroughs inspired by or based on OMSI are reviewed.

### 4.1. Enhancing Catalyst Performance by Tuning the OMSI

When OMSI was evoked and the encapsulation layer formed, metal nanoparticles were restrained on the support, thus they will be prevented from aggregation or leaching during reaction. Tang et al. compared the recycle performance of Au/HAP catalysts calcined at 200 and 500 °C for the selective oxidation of benzyl alcohol [45]. As shown in Figure 7a,b, both conversion and selectivity of the catalyst calcined at 500 °C (Au/H-500) remained unchanged after five cycles of use, and no Au loss was detected. However, the Au/HAP that calcined at 200 °C (Au/H-200) without an encapsulation layer formed has seriously deactivated in the first three reaction cycles, and the used catalyst showed sintered Au nanoparticles and decreased loading. Similarly, reusability of Pd/HAP for Suzuki cross-coupling was significantly enhanced by calcining at 500 °C (Figure 7c) [46]. Moreover, as shown in Figure 7d, Pt/HAP calcined at 500 °C (Pt/HAP-O) exhibited excellent durability under a simulated auto-emission control reaction condition, that the CO oxidation conversion kept unchanged during a 40 h test under 400 °C, which is distinctly better than the Pt/TiO$_2$ reduced at 500 °C (Pt/TiO$_2$-H500, where classical SMSI formed) that deactivated severely in the first 5 h [46]. These evidently demonstrated that the formation of OMSI can effectively inhibit the leaching and aggregation of metal species during reactions in liquid–phase or under elevated temperature with an oxidative atmosphere, which is of great importance for practical application.

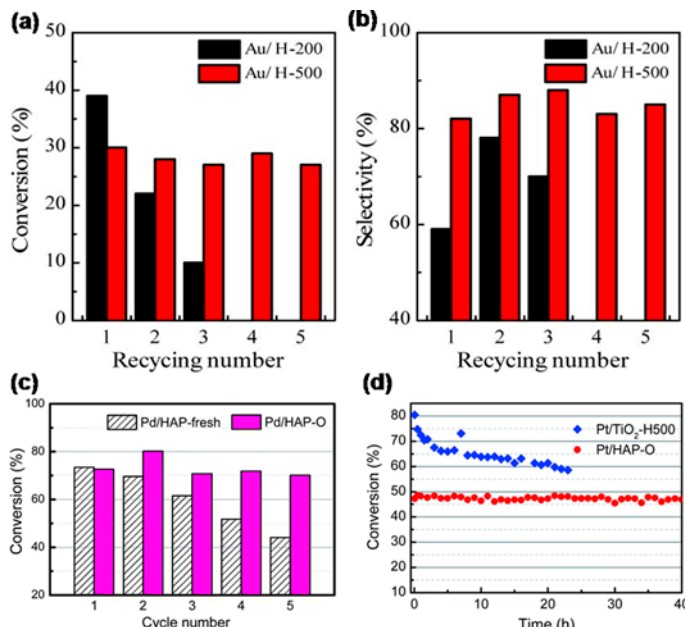

**Figure 7.** (**a**) Conversion and (**b**) selectivity of benzyl alcohol over Au/H-200 and Au/H-500. Reproduced with permission from [45], copyright: 2016 by the American Chemical Society. (**c**) Cycling performance of Pd/HAP-fresh and Pd/HAP-O for the Suzuki cross-coupling reaction. (**d**) CO conversion versus reaction time of Pt/HAP-O and Pt/TiO$_2$-H500 samples at 400 °C. Space velocity of Pt/HAP-O and Pt/TiO$_2$-H500 were ~1,690,000 and 1,100,000 L g$_{Pt}^{-1}$ h$^{-1}$, respectively. Reproduced with permission from [46], copyright: 2018 by The Royal Society of Chemistry.

Catalyst activity mostly depends on the interface sites, which would be impacted by encapsulation when SMSI was evoked. Meanwhile, limited by the support type, catalysts that are able to form SMSI may not be particularly active for certain reactions, although the stability can be enhanced. Controlled SMSI state therefore is needed to obtain catalysts with both improved stability and activity. Commendably, Tang et al. developed an ultrastable Au nanocatalyst with high activity by tuning the OMSI between Au and the composite support TiO$_2$-HAP [47]. As shown in Figure 8, the Au nanoparticle was located at the interfacial regions between the TiO$_2$ and HAP. By high-temperature oxidative treatment, part of the

Au nanoparticle will be covered by HAP and thus obtain excellent sintering resistance, while the part in contact with $TiO_2$ will still be exposed, thus providing interfacial active sites which account for the enhanced activity. The Au/$TiO_2$-HAP calcined at 800 °C (Au/TH-800 in Figure 8) exhibited a much better durability than the commercial three way catalyst (JM888) in a simulated practical testing for 25 days, which is of high potential for practical applications. This work unambiguously highlighted the advantage of designing and tuning OMSI in developing catalysts with enhanced performance.

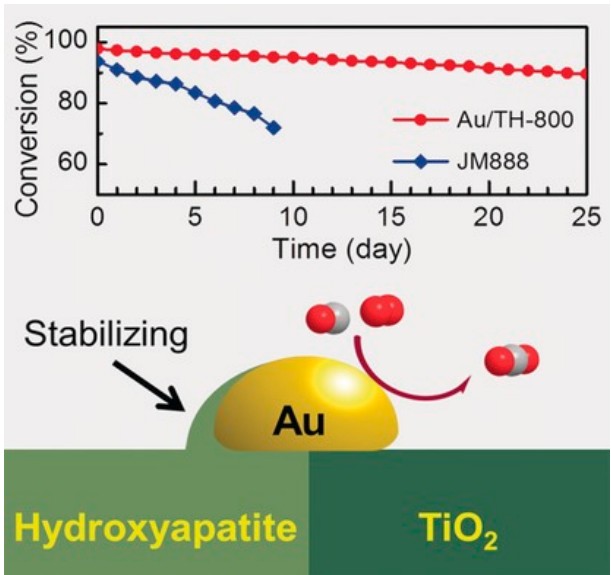

**Figure 8.** Schematic illustration of $TiO_2$-HAP supported Au nanocatalyst with high stability owing to the OMSI between Au and HAP with $TiO_2$, where the upper part shows CO conversion versus reaction time of the Au/TH-800 and commercial three way catalyst JM888 at 400 °C. Reproduced with permission from [47], copyright: 2016 John Wiley and Sons.

*4.2. Discoveries Inspired by or Based on OMSI*

The discoveries of OMSI in the oxide- or nonoxide-supported Au catalysts demonstrate that Au is not an inert metal when forming an SMSI-like effect, raising the question of whether classical SMSI would form, or what would occur, in reducible oxide-supported Au catalysts. The earlier surface science studies based on the model catalysts have not evidenced classical SMSI between $TiO_2$ and Au, which, however, cannot be a practical basis for the assertion that SMSI would not form in Au/$TiO_2$. In fact, the experimental inference based on the real environment was imperative. Inspired by this, Tang et al. synthesized $TiO_2$-supported Au nanocatalysts and further investigated their adsorption performance, mass transport, and electronic state changes under high-temperature reduction, unequivocally uncovering a classical SMSI in Au/$TiO_2$ (Figure 9), and further extended to other reducible oxide (such as $CeO_2$ and $Fe_3O_4$)-supported Au and $TiO_2$-supported IB group metal (Ag and Cu) catalysts [48]. This work is a pivotal breakthrough in promoting SMSI research that verified the universality of the classical SMSI phenomenon and complemented the catalyst system of metal–support interactions (Figure 10). Based on this work, more recently, the size-dependency of classical SMSI was further revealed in Au/$TiO_2$ nanocatalysts, where larger Au particles are more prone to be encapsulated than smaller ones, which brings an in-depth understanding of the SMSI phenomenon and provides a new approach to refine catalyst performance [50].

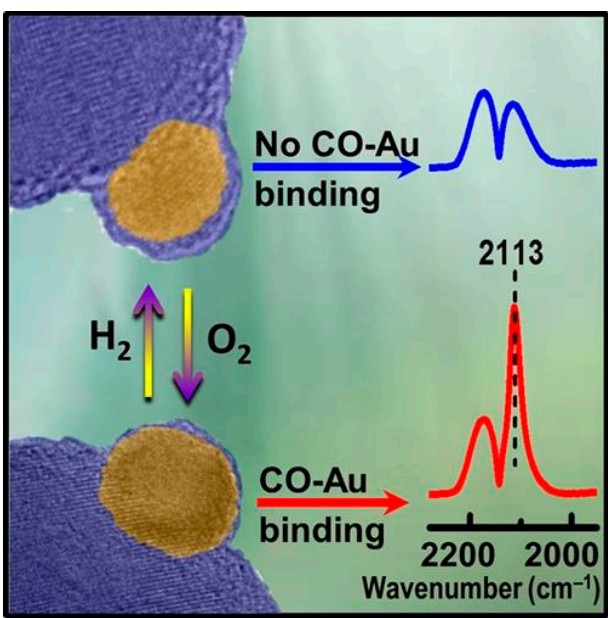

**Figure 9.** Schematic illustration of the classical SMSI phenomenon in Au/TiO$_2$. Reproduced with permission from [48], copyright: 2017 by the American Association for the Advancement of Science.

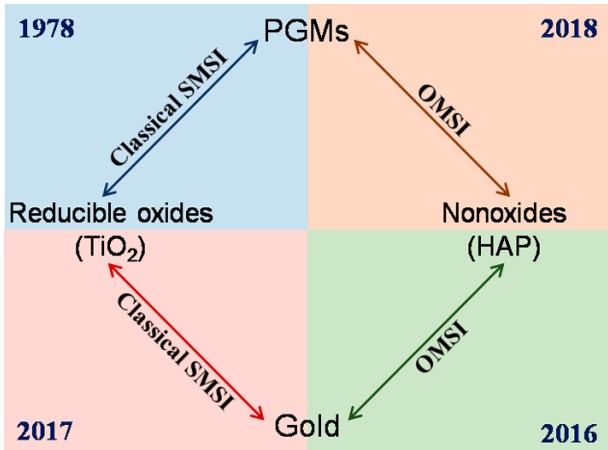

**Figure 10.** Schematic illustration of the catalyst systems for classical SMSI and OMSI.

Contemporaneous with the discovery of the classical SMSI effect in Au/TiO$_2$, a OMSI-like phenomenon between Au and layered double oxide (LDO)-support was reported [49]. Mg-Al layered double hydroxide was employed to support Au nanoparticles and will be dehydrated into LDO when calcined under N$_2$, which will encapsulate Au nanoparticles and accompanied by electron transfer and suppressed adsorption. The Au/LDO exhibited superior anti-sintering stability for CO oxidation and ethanol dehydrogenation at high-temperature due to the SMSI effect. Furthermore, SMSI phenomena that induced by adsorbate or reactant during reaction were reported successively. Christopher's group [76] found that, during the CO$_2$ hydrogenation reaction (at 150–300 °C under 20CO$_2$/2H$_2$ atmosphere), the generated adsorbates (HCO$_x$) strongly bound on the oxide supports (TiO$_2$ and Nb$_2$O$_5$), thus forming HCO$_x$-functionalized oxides with oxygen vacancy which will encapsulate the supported Rh nanoparticles. This new type of SMSI phenomenon involving absorbates was named absorbate-mediated SMSI. Different from classical SMSI, the thin, amorphous, and stable encapsulation layer induced by absorbate-mediated SMSI was permeable to reactants, which significantly increased the selectivity of CO generation. Subsequently, similar absorbate-mediated SMSI was activated to improve the activity and stability of the Cu/CeO$_2$ catalyst for the water–gas shift reaction [77]. Dong et al.

reported a dynamic and reversible SMSI effect induced by carbonization in metal/carbide catalysts (Figure 11) [78]. They found that, by high-temperature treatment under $CH_4/H_2$ atmosphere, the $Au/MoO_3$ nanocatalyst was carbonized to $Au/MoC_x$, where the Au dispersion significantly increased and the charge transfer between Au and $MoC_x$ occurred. Following re-oxidation, $MoC_x$ reversed to $MoO_3$ and the highly dispersed Au aggregated to nanoparticles. This carbonization-activated SMSI tremendously enhanced the low-temperature water–gas shift reaction activity of $Au/MoC_x$ catalysts.

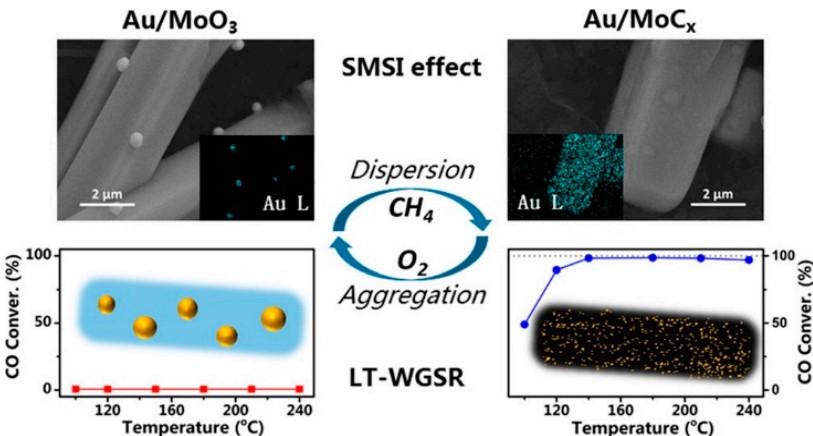

**Figure 11.** Illustration of the SMSI effect between Au overlayers and carbide supports. Reproduced with permission from [78], copyright: 2018 by the American Chemical Society.

More recently, OMSI-like phenomena were discovered for the inert hexagonal boron nitride (h-BN) supported Ni, Fe, Co, and Ru nanocatalysts and the MgO-supported Au nanocatalyst, where a weak oxidizing environment ($CO_2$ and $H_2O$ for Ni/h-BN, and $CO_2$ for Au/MgO) is needed [74,79]. It was found that during the methane dry reforming reaction (at elevated temperatures) by Ni/h-BN, $CO_2$ and/or $H_2O$ will induce the encapsulation of Ni nanoparticles by $BO_x$ overlayers originating from the oxidative etching of the h-BN support, thus forming $Ni@BO_x$ nanostructures providing high activity and stability for the reaction [79]. Similarly, in Au/MgO, the support structure was modified by $CO_2$ treatment, inducing the migration of MgO onto Au nanoparticles and forming a thin encapsulation layer, which is permeable, stable, and water tolerant, exhibiting sinter resistance and stable CO oxidation activity [74]. These OMSI-like phenomena that are induced by the reactant medium for irreducible support-based catalysts further broadened the concept of SMSIs and stimulated new thinking in a deeper understanding of SMSI and in design of highly active catalysts.

## 5. Conclusions and Prospects

Since the first discovery of the oxidation-induced SMSI effect in Au-based catalysts, this phenomenon has been verified in multiple catalyst systems, and the concept of OMSI thus developed rapidly. In this review, we present a comprehensive summary of the history, definition, features, characterization, application, and influence of the OMSI effect, involving related works that have been reported so far. Although being a phenomenon derived from SMSI, OMSI has not only complemented the conventional SMSI systems but also broken through the stereotyped cognitions about the inducing conditions and catalyst systems of SMSI in the past, pioneering the disclosure of various new types of SMSI phenomena and promoting the renaissance of the modern research in metal–support interactions. On the other hand, OMSI provides more potential opportunities for applying metal–support interactions under specific conditions, which has practical significance since a large number of heterogeneous reactions are carried out under oxidative conditions. Many attempts will be invested in further exploring the promising potentials of OMSI, and the future research could be focused on several aspects as specifically proposed here.

The nature and origination of OMSI is the first need to be clarified. In fact, plenty of studies have aimed to reveal the formation mechanism of classical SMSI for decades; however, neither consensus nor a full understanding has been achieved so far. The discovery of OMSI stimulated a reexamination of the conventional SMSI theory, thus it is imperative to unravel the mechanism and driving force of OMSI phenomenon which would be an important part of the constantly refreshed cognition on SMSI. However, the current progress in this is limited. In the previous study, it was found that OMSI in HAP-supported Pt-group metals was also able to occur at high-temperature treatment in inert gas and was partially reversed by water treatment, thus dehydroxylation/dehydration during high-temperature treatments was proposed to possibly induce the formation of OMSI [46]. However, this is not an exclusive conclusion and may not be applicable in other catalysts. Detailed and systemic study is inevitable to clarify the origin of OMSI for each catalyst. Moreover, in addition to the minimization of surface energy, the origination of OMSI should be distinguished from that of classical SMSI. For the classical SMSI, dissociation of molecular hydrogen on the metal surface that initiates the reduction of the support has also been considered as a potential cause. However, for OMSI the oxidative or inert conditions may inspire other possible reasons for its formation, such as the reaction medium modified support deformation [49,74,76,79], the interface reconstruction, or the formation of solid solution. In addition, the causal relationship between the encapsulation reaction and charge transfer is unclear. In the study of classical SMSI, it was reported that the electronic effect was prerequisite for mass transfer [33], but it was also inferred that the physical blocking and electronic effect contribute to the formation of SMSI simultaneously [19]. This is also a difficult issue in OMSI, and fabricating special catalyst systems that can distinguish the two may be helpful.

On the other hand, the conditions for the induction of OMSI should be improved or renovated. So far, the verified OMSIs are generally formed under high temperature. Whether OMSI would be evoked by milder conditions needs to be unraveled and is important for the universality of OMSI application. It has been reported that SMSI can occur in $Pt/TiO_2$ by using $NaBH_4$ or HCHO as a reducing agent at room temperature in the liquid phase, and the formed oxygen vacancies and negatively charged Pt promote the improvement of the toluene oxidation activity [80]. Developing alternative inducing medium and environment for OMSI are also highly expected. Furthermore, unlike classical SMSI, in that a reducible support is prerequisite, the boundary of the catalyst type in which OMSI can form has not been delineated at present. Therefore, generalizing OMSI to diverse catalyst systems is encouraged. Recent discoveries of the OMSI-like phenomena in the inert h-BN and irreducible MgO based nanocatalysts under weak oxidizing environment have evidenced that OMSI is potentially applicable in a variety of catalyst types [74,79]. Moreover, based on the recent discoveries of the SMSI-related phenomena that have been induced by $CO_2$, $CH_4$, and $H_2O$, it can be predicted that the reactant medium-induced OMSI phenomena would become a hot point as they are closely related to the research on catalyst changes during reaction.

In addition to the above aspects that need attention in the OMSI research, exploiting the OMSI effect to develop more strategies for tuning catalyst performance and expanding its application fields are certainly significant too. Beyond thermocatalysis, it is worthwhile to attempt to apply the OMSI-based control strategies to electrocatalysis and photocatalysis. We look forward more discoveries and applications derived from OMSI in the future.

**Author Contributions:** Conceptualization, B.Q.; writing—original draft preparation, X.D.; figures, X.D.; literature research, X.D. and H.T.; writing—review and editing, H.T. and B.Q.; supervision B.Q.; project administration, B.Q. All authors have read and agreed to the published version of the manuscript.

**Funding:** This work was funded by Natural Science Foundation of China and Japan Society for the Promotion of Science Cooperative Research Project (21961142006), National Natural Science Foundation of China (21972135, 21776270, 21902040), LiaoNing Revitalization Talents Program (XLYC1807068), and DNL Cooperation Fund, CAS (180403). H.T. acknowledges the support from Natural Science Foundation of Hebei Province (B2019201158), and the Advanced Talents Incubation Program of the Hebei University (521000981212). X.D. thanks the support by China Postdoctoral Science Foundation (2018M641725).

**Data Availability Statement:** Not applicable.

**Conflicts of Interest:** The authors declare no conflict of interest.

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
