# Peer review of "Oxidative Strong Metal–Support Interactions"

_catalysts, doi:10.3390/catal11080896_

Round 1
Reviewer 1 Report
This manuscript concerns with comprehensive review article under the title “Oxidative Strong Metal-Support Interactions”. It is certainly a good effort to summarize a brief history as well as OMSI and its difference from classical SMSI. The identification of OMSI and the corresponding characterization methods are discussed. The application of OMSI in enhancing catalyst performance, and the influence of OMSI with new types of SMSI, are discussed. This manuscript should be accepted after minor revision as the concept and overall writing is beneficial for the scientific community and industry.
Herein, I am summarizing my concerns.
- Some figures are with low resolution and suggested to use a better image especially, figure 3 and 6.
- The author should consider the plagiarism and should be reduced to an acceptable range. Currently the similarity report shows 18% and some parts are just copy and paste. If there is no copyright issues than can be proceed.
- The author used very old references throughout the manuscript although a lot of work and review articles are published in last few years. It is suggested to update references with new information to help the scientific society as we are living in year 2021.
- If possible, it is suggested to prepare a graphical abstract to understand the general idea of the review to attract more audience.

Author Response
Response to the comments and questions of reviewer 1
Manuscript ID: catalysts-1306795
Title: Oxidative Strong Metal-Support Interactions
Author: Xiaorui Du et al
We appreciate the reviewer's encouraging comments and constructive suggestions which are very helpful in improving our manuscript. Specific responses to each suggestion are listed below.
Reviewer 1
This manuscript concerns with comprehensive review article under the title “Oxidative Strong Metal-Support Interactions”. It is certainly a good effort to summarize a brief history as well as OMSI and its difference from classical SMSI. The identification of OMSI and the corresponding characterization methods are discussed. The application of OMSI in enhancing catalyst performance, and the influence of OMSI with new types of SMSI, are discussed. This manuscript should be accepted after minor revision as the concept and overall writing is beneficial for the scientific community and industry.
Herein, I am summarizing my concerns.
- Some figures are with low resolution and suggested to use a better image especially, figure 3 and 6.
Response:
Figure 3 and 6 have been revised with higher resolution.
- The author should consider the plagiarism and should be reduced to an acceptable range. Currently the similarity report shows 18% and some parts are just copy and paste. If there is no copyright issues than can be proceed.
Response:
Thank you so much for the suggestion, the whole text has been accordingly revised.
- The author used very old references throughout the manuscript although a lot of work and review articles are published in last few years. It is suggested to update references with new information to help the scientific society as we are living in year 2021.
Response:
Thanks for the valuable suggestion. Since the history of SMSI and OMSI were described in the manuscript, earlier references involving the important discoveries and reports during were cited inevitably. As reminded by reviewer, we have re-examined the references and cited some important relevant researches in the past three years.
- If possible, it is suggested to prepare a graphical abstract to understand the general idea of the review to attract more audience.
Response:
Thanks for the advice. A graphical abstract was provided as follows.
Reviewer 2 Report
This is an interesting and valuable review of OMSI. For this reader it did meet the hope of authors: “… raise interest for researchers and inspire further attempts to explore more potential of OMSI in heterogeneous catalysis.” There are several grammatical errors throughout the manuscript.
- Introduction
This section is a concise and comprehensive coverage of the background to OMSI. This is a new phenomenon for me and the acronyms (SMSI, HTR, PGMs, NPs, classical SMSI, OMSI) used in the Introduction initially caused some confusion. I think a little restructuring, with perhaps a reduction in acronyms (only classical SMSI and OMSI are frequently used throughout the manuscript), so that it is more readily accessible to all readers would be helpful. The following wording is unclear: “… mass transport that the reduced support encapsulating metal particles”.
- Definition, features and catalyst systems of OMSI
This section is also concise providing more specific details of OMSI and compared to classical SMSI – nicely summarized in Table 1. More acronyms are introduced (HAP, HTO), which seem unnecessary. The four characteristics of OMSI are analogous to the four characteristics for classical SMSI.
- Identification and characterization of OMSI
This section provides comprehensive details of the methods used to test the four characteristics of OMSI: adsorption behaviour, mass transport, electron transfer and the effect on these three characteristics after subsequent reduction.
- Application and influence of OMSI
This section discusses applications and advantages of OMSI over classical SMSI. Subsections are enhancing catalyst performance and discoveries.
- Conclusions and prospects
This final section is a detailed summary of OMSI and the first four sections of the manuscript, as well as the potential of OMSI. Some statements are referenced and others not. This section appears to repeat the information from previous sections and introduce new information. That is, I think this section should only be a summary of the discussion.
Author Response
Response to the comments and questions of reviewer 2
Manuscript ID: catalysts-1306795
Title: Oxidative Strong Metal-Support Interactions
Author: Xiaorui Du et al
We appreciate the reviewer's detailed comments and constructive suggestions which are very helpful in improving our manuscript. We have checked the grammar and revised advisably to make the expression more clear. Specific responses to each suggestion are listed below.
Reviewer 2
This is an interesting and valuable review of OMSI. For this reader it did meet the hope of authors: “… raise interest for researchers and inspire further attempts to explore more potential of OMSI in heterogeneous catalysis.” There are several grammatical errors throughout the manuscript.
- Introduction
This section is a concise and comprehensive coverage of the background to OMSI. This is a new phenomenon for me and the acronyms (SMSI, HTR, PGMs, NPs, classical SMSI, OMSI) used in the Introduction initially caused some confusion. I think a little restructuring, with perhaps a reduction in acronyms (only classical SMSI and OMSI are frequently used throughout the manuscript), so that it is more readily accessible to all readers would be helpful. The following wording is unclear: “… mass transport that the reduced support encapsulating metal particles”.
Response:
Thanks for the point. We used these acronyms to avoid frequent appearance of long nouns that commonly used. However, as reminded by the reviewer, much acronyms may cause confusion. After consideration, the acronyms PGMs (Pt-group metals), NPs (nanoparticles), HTR (high-temperature reduction), HTO (high-temperature oxidation), and A-SMSI (absorbates-mediated SMSI) have all been revised.
The sentence "… mass transport that the reduced support encapsulating metal particles" has been revised as "... mass transport that metal particles will be encapsulated by the reduced support".
- Definition, features and catalyst systems of OMSI
This section is also concise providing more specific details of OMSI and compared to classical SMSI – nicely summarized in Table 1. More acronyms are introduced (HAP, HTO), which seem unnecessary. The four characteristics of OMSI are analogous to the four characteristics for classical SMSI.
Response:
The acronym HTO (high-temperature oxidation) has been revised. The acronym HAP refers to hydroxyapatite which was frequently used in the article to refer certain catalyst such as Au/HAP (hydroxyapatite supported Au catalyst). Therefore, HAP was retained.
In fact, OMSI is one of the typical phenomena that derived from SMSI, and its features are resemble with that of classical SMSI. To illustrate the history background of OMSI, the features of SMSI were clearly described firstly. To provide a comprehensive definition and identification criteria of OMSI, its characteristics were detailed listed, which we think are necessary.
- Identification and characterization of OMSI
This section provides comprehensive details of the methods used to test the four characteristics of OMSI: adsorption behaviour, mass transport, electron transfer and the effect on these three characteristics after subsequent reduction.
Response:
Thanks for the comment.
- Application and influence of OMSI
This section discusses applications and advantages of OMSI over classical SMSI. Subsections are enhancing catalyst performance and discoveries.
Response:
Thanks for the comment.
- Conclusions and prospects
This final section is a detailed summary of OMSI and the first four sections of the manuscript, as well as the potential of OMSI. Some statements are referenced and others not. This section appears to repeat the information from previous sections and introduce new information. That is, I think this section should only be a summary of the discussion.
Response:
Thanks for the suggestion.
In fact, the first paragraph of this Section is a summary of the discussion, in which also emphasized the practical significance of OMSI to the study of metal-support interactions and to the catalysts development. In addition to the current understanding, we also proposed some unsolved problems as the possible future research aspects on OMSI, such as revealing the nature and origination of OMSI, renovating the conditions for the induction of OMSI, digging more catalyst systems for OMSI, and exploiting OMSI effect to develop more strategies for tuning catalyst performance. These points of view were described in the last three paragraph of this Section, which we think is substantial help in raising interest for researchers and inspiring further discoveries and applications about OMSI in heterogeneous catalysis. Therefore, the main content of this Section was retained.
